# Neuropsychopharmacological Induction of (Lucid) Dreams: A Narrative Review

**DOI:** 10.3390/brainsci14050426

**Published:** 2024-04-25

**Authors:** Abel A. Oldoni, André D. Bacchi, Fúlvio R. Mendes, Paula A. Tiba, Sérgio Mota-Rolim

**Affiliations:** 1Center for Mathematics, Computing and Cognition, Federal University of ABC, São Bernardo do Campo 09606-045, Brazil; abel.o@aluno.ufabc.edu.br (A.A.O.); paula.tiba@ufabc.edu.br (P.A.T.); 2Faculty of Health Sciences, Federal University of Rondonópolis, Rondonópolis 78736-900, Brazil; bacchi@ufr.edu.br; 3Center for Natural and Human Sciences, Federal University of ABC, São Bernardo do Campo 09606-045, Brazil; fulvio.mendes@ufabc.edu.br; 4Brain Institute, Federal University of Rio Grande do Norte, Natal 59078-970, Brazil

**Keywords:** dream recall, lucid dreaming, REM sleep, metacognition, self-consciousness, dopamine, acetylcholine, galantamine, acetylcholinesterase inhibitors, sesquiterpene lactones

## Abstract

Lucid dreaming (LD) is a physiological state of consciousness that occurs when dreamers become aware that they are dreaming, and may also control the oneiric content. In the general population, LD is spontaneously rare; thus, there is great interest in its induction. Here, we aim to review the literature on neuropsychopharmacological induction of LD. First, we describe the circadian and homeostatic processes of sleep regulation and the mechanisms that control REM sleep with a focus on neurotransmission systems. We then discuss the neurophysiology and phenomenology of LD to understand the main cortical oscillations and brain areas involved in the emergence of lucidity during REM sleep. Finally, we review possible exogenous substances—including natural plants and artificial drugs—that increase metacognition, REM sleep, and/or dream recall, thus with the potential to induce LD. We found that the main candidates are substances that increase cholinergic and/or dopaminergic transmission, such as galantamine. However, the main limitation of this technique is the complexity of these neurotransmitter systems, which challenges interpreting results in a simple way. We conclude that, despite these promising substances, more research is necessary to find a reliable way to pharmacologically induce LD.

## 1. Introduction

Dreams are characterized by perceptions, emotions, and cognitions that happen in any sleep stage. Lucid dreaming (LD) is a spontaneously rare type of dream, mainly associated with REM sleep, in which subjects recognize that they are dreaming. In this article, we review studies on substances derived from plants and drugs used to intensify oneiric activity and induce LD. We first describe the processes that regulate sleep and the main neurotransmitters that control REM sleep (see Appendix A). Next, we discuss the neurophysiology and phenomenology of LD to finally review substances that increase metacognition, REM sleep duration, and dream recall frequency, thus with the potential to induce LD.

## 2. Lucid Dreams: Definition and Neurophysiology

LD occurs when dreamers know they are dreaming; that is, they comprehend that the reality they are experiencing is not the waking one, but the sleeping one. Although LD is spontaneously rare in the general population, it is a learnable skill. LaBerge [1] underwent self-training for three years using cognitive tasks such as self-suggestion with a focus on his motivation and intention to stay lucid in the next dream he would have. Refining this training, he developed the technique mnemonic induction of lucid dreaming (MILD), which consists of remembering to perform future actions, using mental visualization or verbalization about the act that one would like to perform (e.g., “I will remember to recognize that I am in a dream while dreaming”).

The objective verification of LD can occur in a real-time experimental setting, through communication by eye movements (which do not present complete muscle atonia as other body muscles) during rapid-eye-movement (REM) sleep [2]. By combining these movements before falling asleep, participants, when becoming lucid in the dream, can perform pre-agreed eye movements (e.g., left and right), which can be detected in the electrooculogram [3].

Although LD happens predominantly during REM sleep, it can also occur during sleep onset (N1) and light sleep (N2) stages [4,5]. Eye-signaling reports of LD in the deep sleep (N3) stage have not yet been found. However, practitioners of Transcendental Meditation report being able to maintain self-awareness during this deep sleep stage [6,7]. In addition, Yoga Nidra practitioners report the possibility of cultivating self-awareness throughout the entirety of sleep [8,9]. Physiological studies of LD substantially focus on the occurrence of lucid REM dreams, which are more easily verified. For convenience and based on the available literature, the focus of the description of LD physiology here will be exclusively on the phenomenon during REM sleep.

Using a combination of electroencephalography and functional magnetic resonance imaging, Dresler et al. [10] observed activation in neocortical and cortical networks during lucid REM sleep, mainly in the precuneus, occipitotemporal cortex, frontopolar cortex, and right dorsolateral prefrontal cortex. Among possible neurophysiological and phenomenological mechanisms related to LD, it is suggested that the increased activity in the right dorsolateral prefrontal cortex may be related to self-centered metacognitive evaluation and, together with the parietal lobules, corresponds to working memory activation. Frontopolar areas are involved in the processing of internal states, emotions, and thoughts, which have been widely reported in LD. The higher difference in activation between lucid and non-lucid REM sleep was in the precuneus, which is related to self-referential processing, i.e., first-person perspective. The exceptional clarity of scenery (vividness) in LD [11] coincides with the activation of occipitotemporal cortices, which are ventral parts of visual processing and are involved in various aspects of conscious visual perception [10].

Low metacognition in non-lucid dreams (exemplified by a lack of volitional capacity, impaired critical thinking, and a lack of self-reflection on one’s state) can be explained by the deactivation of the dorsolateral prefrontal cortex and the frontopolar cortex during REM sleep [12]. To test whether these areas are active during LD, Filevich et al. [13] conducted a study with thought monitoring tasks, metacognition questionnaires, and neuroimaging. When separating participants into high- and low-lucidity groups, the former showed increased gray matter volume in regions BA9, especially in the dorsolateral prefrontal cortex, and BA10, specifically in the frontopolar cortex, following the authors’ hypothesis. The BA10 area has been associated with visual and metacognitive monitoring abilities. Additionally, greater activity was measured in these areas during metacognitive tasks compared to a control condition that did not require these abilities [13].

Another study [14] evaluated the neuroanatomy of frequent lucid dreamers’ traits. The authors showed significantly increased functional connectivity at rest between the left anterior prefrontal cortex and bilateral angular gyrus, bilateral middle temporal gyrus, and right inferior frontal gyrus. In contrast to the findings of Filevich and colleagues, there was no difference in gray matter density between the two groups. Among the areas differentiated between LD and normal dreams, there was an increase in functional connectivity between the frontopolar cortex and temporoparietal associative areas, with overlap of the frontoparietal network. The authors conclude that the changes observed in the functional connectivity of the anterior prefrontal cortex may underline metacognitive judgments and functions that are presented in LD (Figure 1). They also suggest future studies involving cholinergic modulation, since pro-cholinergic drugs tend to increase frontoparietal activity, as will be detailed in the next section.

## 3. Pharmacological Induction of Lucid Dreams

Since spontaneous LD is rare, an effective method for its induction is currently desired, both for facilitating experimental studies and for understanding the neural basis of metacognition and dream self-awareness [16]. In this section, we first describe cholinergic and dopaminergic substances, which are the main candidates to induce LD. Then, we review some case reports of less-studied substances. Finally, potential natural candidates that increase dream vividness and/or recall, such as plants and herbs, are discussed.

### 3.1. Cholinergic Substances

Acetylcholinesterase inhibitors (AChEIs) constrain the enzyme that hydrolyzes acetylcholine (ACh) and increases its levels in the synapse. Many plants have been used in traditional medicine to treat cognitive deficits, including neurodegenerative disorders. These plants are a rich source of compounds with antioxidant activity and acetylcholinesterase inhibition. Most studies have focused on anticholinesterase alkaloids, but other major classes of compounds reported to have such activity are terpenoids, glycosides, flavonoids, and coumarins [17,18,19,20]. AChEIs, such as donepezil, rivastigmine, and galantamine, despite sharing this central mechanism, bear specific properties that can potentially engender subtle disparities in their effect profiles [21]. Donepezil is a reversible, long-acting, and selective AChEI with no inhibitory effect on butyrylcholinesterase (BuChE). The selectivity of acetylcholinesterase (AChE) over BuChE could hypothetically mitigate side effects associated with BuChE inhibition. Its reversible nature and long-term action may also contribute to its sustained effectiveness throughout the day at a single daily dose [22]. In contrast, rivastigmine, a “pseudo-irreversible” inhibitor of both AChE and BuChE, reversed this inhibitory effect over time. This property may allow for superior control over brain ACh levels, potentially minimizing ACh toxicity. Furthermore, rivastigmine might show selectivity for AChE in the cortex and hippocampus, as opposed to AChE in other brain regions, which could potentially enhance cognitive effectiveness with fewer peripheral side effects [23].

Galantamine is a natural tertiary alkaloid from the Amarylidacea family, originally isolated from the bulbs of Snowdrop and Narcissus species [24] (Figure 2). In contrast to donepezil and rivastigmine, galantamine is not only an AChEI, but also a positive allosteric modulator of nicotinic ACh receptors. This property allows galantamine to potentiate the effects of ACh on these receptors, possibly enhancing its effectiveness in improving cognitive function [25]. This dual mechanism of action could potentially elevate the efficacy of galantamine over other AChEIs that are devoid of this additional action on nicotinic receptors.

There is ample evidence that REM sleep is modulated by ACh (see Appendix A) and AChEIs are involved in the phasic activation and stabilization of REM sleep, which increases the chance of inducing LD [27]. LaBerge’s patent [28] shows that AChEIs, such as galantamine, donepezil, rivastigmine, and huperzine A, significantly increase the clarity of cognition, lucidity, self-reflection, recall, control, bizarreness, and visual vividness, with few side effects. With donepezil, 8 out of 10 subjects experienced LD, and only 1 out of 10 reported LD on placebo night. However, it is important to note that this medication has some undesired side effects, such as nausea. Huperzine A, from *Huperzia serrata*, a potent plant-derived AChEI alkaloid, presents similar effects, but with a much higher dose compared to donepezil. Rivastigmine and galantamine had the same effects on lucidity, with fewer side effects. LaBerge proposes that in addition to AChEIs, other cholinergic agonists, such as muscarinic receptor agonists, or presynaptic receptor antagonists might induce LD.

More recently, LaBerge et al. [29] investigated whether galantamine would induce LD. In a final sample of 121 individuals, 3 doses of galantamine were distributed to be used over 3 days: 0 mg (control condition), 4 mg, and 8 mg. Galantamine was found to be a dose-dependent inducer of LD, with few or no side effects. There was also an increase in dream recall, sensory vividness, bizarreness, and dream complexity, as well as a decrease in negative emotions. Participants with previous experience with LD were more likely to experience LD, and of the 10 individuals who had never experienced LD, 4 of them reported one lucid dream episode with the higher dose.

Another study combined galantamine with two cognitive strategies: meditation and dream reliving (MDR) and wake back to bed (WBTB) [30]. MDR is a technique for reliving dreams with critical self-reflection; while the events are happening in mental imagery, the person tries potential ways to become lucid, like scenarios of nightmares. The meditation used involves practicing self-awareness and being non-reactive to stressful oneiric content. WBTB is an LD induction technique in which the subject wakes up late at night when REM sleep and vivid dreams are more prone to happen. Then, the person goes back to sleep, potentially experiencing the same dream, or having vivid dreams, augmenting the probability of achieving lucidity. The study used 35 participants, and 8 nights of dreaming were recorded. Galantamine outperformed placebo with both MDR and WBTB techniques, which did not differ in terms of lucidity. MDR increased reflexivity, as well as fear and threat, compared to WBTB. As the number of lucid dreams achieved was not mentioned in the article, Baird et al. [15] contacted the authors, and the results showed that 9% of participants reported LD in the WBTB + placebo condition and 11% in the MDR + placebo condition, while 40% of participants reported LD in the WBTB + galantamine condition, and 34% in the MDR + galantamine condition. These results highlight the potential drug candidate, galantamine, to induce LD. The authors suggest that MDR may bring exposure to trauma or conflict resolution, depending on the dreamer’s intention.

Although AChEIs increase cholinergic activity, it seems that cholinergic agonists are not always effective in LD induction, as is the case of alpha-GPC. Alpha-GPC is a precursor of ACh, which, in contrast to ACh itself, can cross the blood–brain barrier. The procedures adopted in a study with alpha-GPC involved three non-consecutive nights: one with a placebo, one with the supplement, and one wash-out night [31]. Participants’ reports were recorded, and the LuCiD scale was used to measure the level of consciousness in dreams. No inducing effect of this substance was observed; of the 33 participants, six experienced LD: two of the advanced practitioners, two of the novices with alpha-GPC, and two novices on placebo nights. One possible reason is that no mental training was used to achieve lucidity, and cognitive training is widely used in experimental studies [15].

Considering the pharmacological action of galantamine as an AChEI and a positive allosteric modulator of nicotinic ACh receptors, it is plausible to suggest that this unique combination of mechanisms may account for the superior efficacy observed in LD induction. In contrast, despite being AChEIs, donepezil and rivastigmine lack the same action on nicotinic receptors, potentially making them less effective than galantamine at inducing LD. In parallel, alpha-GPC, a cholinergic precursor that can increase available brain ACh levels, lacks the positive allosteric modulation of nicotinic ACh receptors. Furthermore, galantamine prolonged the action of ACh by inhibiting its degradation. In contrast, a-GPC enhanced the availability of choline, an ACh precursor, but did not directly influence ACh degradation. Thus, galantamine’s ability to prolong ACh action may promote LD, and might be in accordance with the high modulation of lucid REM sleep by ACh [27]. In addition, ACh seems to modulate activity on the dorsolateral prefrontal cortex, an executive area that is activated during LD. However, these hypotheses remain rooted in the current understanding of the mechanisms of action of these drugs and the LD phenomenon. Further research is necessary to validate these propositions and better understand the relationship between cholinergic pharmacology and LD.

### 3.2. Dopaminergic Substances

Data obtained from experiments with rats [32] suggest that galantamine increases dopamine (DA) cell firing in the ventral tegmental area, through allosteric potentiation of nicotinic (not muscarinic) receptors. Specifically, it increases extracellular DA levels in the prefrontal cortex, especially the medial prefrontal cortex. The authors note that the cognitive mechanisms behind this effect are still not fully understood. AChEIs might play a role in the dopaminergic system through reward inhibition. Even though the mechanisms are still not fully understood, and different doses might affect this type of behavior, repression of irrelevant stimuli, such as seeking drugs, in both humans and rats seems to be prevalent. Suppressing irrelevant stimuli while enhancing others, as in the case documented by Kjaer et al. [33] during Yoga Nidra, might be a way to become lucid in dreams.

Yoga Nidra is a meditation technique systematized by Swami Satyananda Saraswati, who took the ancient scriptures contained in the Vedantic literature. In his book “Yoga Nidra”, many definitions are made, such as “a technique in which you learn to relax consciously” (p. 4); “the consciousness is in a state between waking and sleep, but it is subject to neither [34]”. This meditation has been gaining attention in recent years, having many benefits such as an increased subjective quality of sleep, as well as treating insomnia; decreasing stress, anxiety, and depression; and increasing psychological well-being [35].

An intriguing study demonstrated that during Yoga Nidra meditation, [11C]-Raclopride, a selective antagonist of D2 and D3 receptors, showed less binding in the ventral striatum compared to the control group when the practitioners of the meditation were at rest [33]. The radioligand had 7.9% less biding, which can be interpreted as endogenous DA in the ventral striatum being augmented by 65%. In addition, participants reported less readiness to act and enhanced sensory imagery. Parker et al. [8] argue that the former results only show a preliminary practice of Yoga Nidra, not the practice per se, since there are definitions of Yoga Nidra that take into account that there should be increased delta waves in the brain, with the subject simultaneously conscious. Indeed, a recent study has shown electrophysiological evidence of meditation-naïve participants in a 2-week intervention of Yoga Nidra, showing local sleep—that is, increased synchronization evidenced by delta waves—in the central area and reduction in the prefrontal area. This result was accompanied by enhanced subjective sleep quality and efficiency [36]. Even though delta waves might indicate loss of consciousness, Yoga Nidra shows the contrary. A similar phenomenon is the paradoxical pharmacological dissociation, defined as drug-induced states that enhance delta oscillations but preserve consciousness [37], which will be further commented on in the general discussion (see Section 5).

Yoga Nidra, as well as Dream Yoga (from Tibet), regard LD just as a step that could be facilitated by entering consciously into sleep. With this in mind, more neuroimaging studies regarding self-consciousness during sleep should be carried out to make solid conclusions on the subject, since being lucid in sleep might not be exclusively associated with executive control. Nevertheless, it can be hypothesized from the studies above that DA is involved in self-consciousness. Indeed, DA might play a crucial role in self-awareness. A causal role of the medial prefrontal cortex and medial parietal regions might be speculated. These regions have connections to the so-called default mode network (Figure 3), such as the angular gyrus, insula, striatum, and thalamus, where DA interacts with GABA-induced synchronized gamma oscillations [38].

A study found that synchronous transcranial alternating-current stimulation in theta frequency applied to the frontoparietal network improved working memory performance when cognitive demands were high. The frontoparietal network, and working memory ability, might be crucially involved in LD [14,15].

In line with the evidence that DA is involved in metacognition, becoming lucid in dreams may involve conscious self-monitoring, and DA could play a crucial role in this process. Joensson et al. [40] aimed to investigate whether dopaminergic stimulation could enhance self-awareness, metacognition, and self-monitoring by administering L-dopa, a dopamine precursor, to healthy individuals. The results demonstrate that self-referential processing improved, leading to increased self-awareness and self-monitoring. In addition, individuals who took L-dopa exhibited improved performance compared with the control group. Thus, the subjective experience of consciousness became a more accurate predictor of performance. To further examine the neural basis of these improvements, the authors used magnetoencephalography to analyze brain regions implicated in self-monitoring. They found that dopaminergic stimulation increased activity in the medial prefrontal cortex, a region that may exhibit differential dopaminergic modulation during LD due to its association with conscious self-monitoring.

L-dopa is the natural amino acid precursor of DA [41] and is used in the treatment of Parkinson’s disease, but it can have behavioral and cognitive side effects. Preliminary results show that this dopaminergic medication has positive effects on metacognition in Parkinson’s disease patients [42], but it seems that the drug can cause vivid dreams in these patients [43,44,45,46]. REM sleep behavior disorder might contribute to the vivid dreams attributed to Parkinson’s disease patients [39]. Occurring in approximately one-third of patients with Parkinson’s disease, REM sleep behavior disorder is characterized by a loss of muscle atonia during REM, and is associated with aggressive and vivid dream content in Parkinson’s disease patients [47]. Excluding Parkinson’s disease patients with REM sleep behavior disorder, it was found that DA agonist dosage in the treatment of Parkinson’s disease made dreams less vivid and less emotional [48]. It was also observed that visual vividness was related to amygdala volume and the thickness of the medial prefrontal cortex. Other drugs that can treat Parkinson’s disease such as pramipexole, a non-ergot dopamine D2/D3 receptor agonist, seem to produce vivid dreams as side effects [49]. It is worth mentioning that, in normal individuals, it inhibits DA synthesis and release, but when DA neurons are impaired—as in the case of Parkinson’s disease—the drug acts as a potent postsynaptic dopamine D2 receptor agonist.

Taking these findings together, it is plausible to infer a causal role of DA in the generation of dreams, having an impact on oneiric emotional content and vividness. Both factors, when enhanced, increase the possibility of the dreamer to understand the narrative and become lucid. Knowing that DA is also related to metacognition, self-awareness, and suppressing irrelevant stimuli, future studies should address the role of DA in inducing vivid dreams and helping one become self-conscious when suppressing stimuli that are barriers to understanding the nature of the oneiric reality the dreamer is immersed in.

### 3.3. Case Reports with Other Substances

Haas et al. [50] report a case of a 65-year-old woman diagnosed with multiple myeloma, with a history of chronic pain. On the second day after her admission to the hospital, she complained of LD, which were terrifying (see lucid nightmare in the general discussion). The authors concluded that the cause was pregabalin, since upon discontinuing the medication, LD also ceased. Pregabalin is used to treat neuropathic pain; however, it has various side effects. It is a 3-isobutyl derivative of GABA, and the mechanisms of action of pregabalin are associated with its strong binding to alpha-2-delta sites, which are a subunit of voltage-dependent calcium channels in central nervous system tissues that regulate calcium influx into nerve terminals. This reduces the release of excitatory neurotransmitters including NA, glutamate, DA, 5-HT, and substance P.

In another case report, Mousailidis et al. [51] describe a patient who experienced visual hallucinations and agitation associated with an increase in the dose of pregabalin, which completely resolved after discontinuation of the drug. The manufacturer reports abnormal dreams as an infrequent characteristic of the drug’s use.

In a recent study, an online survey was conducted on food and supplements that could affect nocturnal consciousness (e.g., dream recall, LD, and hypnagogic hallucinations) [52]. Partial correlations showed that taking vitamin supplements increases either dream recall or LD frequency. In addition, vitamin intake had a positive correlation with hypnagogic recall. The surprising result was that the consumption of fish was positively correlated with LD. The author speculates that this result might be explained by the omega-3 fatty acids that have already been suggested to underlie some cognitive abilities. However, these results should be taken with caution, since the individuals could be training LD or compensating for a deficiency in omega-3. Moreover, taking antidepressants positively correlated with LD, which might be linked to the REM sleep rebound caused by substances that affect the serotonin (5-HT) system. Another interesting result was that eating chili was correlated with hypnagogic recall, and this could be due to the capsaicin substance found in chili, which has already been linked to cognitive functioning, as the author mentions. These results could be promising for the intake of substances to induce LD. Although the main limitation of this research is that it is an online survey, and hence prone to subjectivity, the author emphasizes that field research could be fruitful in accessing variables related to sleep, dreaming, and LD.

Sergio [53] conducted a self-case study and began testing substances that stimulate the brain. 2-Dimethylaminoethanol (DMAE) proved to be positive in his personal experiences to induce LD. It is also described as a powerful stimulant of brain reticular formation, causing excitement and consciousness during REM sleep, and shortening the amount of sleep needed each night, promoting a sense of mental clarity upon waking. As a result, DMAE helps a person stay at a higher level of consciousness while dreaming, which facilitates the realization that one is dreaming. DMAE is a relatively non-toxic compound with vitamin-like properties similar to choline and is converted to choline in the body. They differ in terms of membrane synthesis because choline cannot be transported across cell membranes to the sites where it is needed, while DMAE can easily pass through cell membranes. Sergio advises to use 50 mg per day for 2 weeks to achieve peak effects. In combination, he would wake up 1 h before his usual wake-up time and engage in visualization techniques to become lucid, allowing him to enter directly into LD.

DMAE (Deanol) is a classical nootropic [54] found in many nutritional supplements, such as salt of tartaric acid. Interestingly, it can also be absorbed through eating fish, such as salmon and shellfish, which supports the study of Biehl, cited above, where eating fish was related to LD, and also with the self-experiment cited above where DMAE was shown to induce LD [53]. Delving into its properties as a cognitive booster, there is evidence that DMAE increased choline and ACh extracellular levels in rats, thus improving spatial memory. It has also been shown that, in human studies, DMAE combined with supplements (such as vitamins) improved alertness and attention [54]. Taking these results together, future studies should address whether frequent consumption of fish is actually related to LD.

Richter’s [55] patent presents a nutritional supplement for improving sleep and increasing LD, containing *Calea ternifolia*, L-5-hydroxytryptophan (L-5-HTP), and vinpocetine. The supplement may include secondary ingredients, such as melatonin, Mugwort extract, DMAE, passionflower extract, green tea extract, and wild lettuce extract. It may also contain vitamins B, D, and C, zinc, magnesium, and calcium. Among the reasons why the supplement could enhance LD, the author reports that (1) *Calea ternifolia* is a known dream herb used by the Chontal Natives of Mexico (see the next session for more details); (2) green tea (*Camellia sinensis*) is a well-known beverage that is rich in antioxidants; (3) Mugwort (*Artemisia vulgaris*) extract has been used for medicinal purposes for centuries and is known to cause a dreamlike state of consciousness (it will be mentioned again in the next session); (4) passionflower (*Passiflora* spp.) has mild sedative properties that can be used to treat insomnia and anxiety; (5) wild lettuce is an ingredient in some sleep tonics and has historically been considered a mild sedative; (6) L-5-HTP is known to increase serotonin levels, which, in turn, improves sleep quality; (7) DMAE may increase acetylcholine levels; (8) Vinpocetine (from *Vinca minor*) is known to help memory and improve mental functioning; (9) melatonin has properties that are detailed below; (10) B vitamins are known to regulate the body’s energy processes and promote good health; (11) vitamin C is also a necessary ingredient for maintaining good health; (12) zinc is used by hundreds of enzymes that regulate many of the body’s functions; and (13) vitamin D, calcium, and magnesium are additional elements that are utilized by the body. The author presents two brief case reports of individuals who became lucid and controlled their dreams when taking this supplement.

Melatonin (N-acetil-5-metoxitriptamina) is a hormone produced by the pineal gland that informs an organism that it is night, which for diurnal animals (such as us, humans), represents rest and sleep. In this way, melatonin has been used as a sleep-promoting agent [56,57]. Besides this hypnotic function, it seems that melatonin can also increase dream recall frequency, vividness, and bizarreness [58]. This can be partially explained by better coordination of the sleep stages, especially REM sleep. Despite these results, as far as we know, there are no studies that investigate whether melatonin can be used to induce LD.

Interestingly, a pilot study demonstrated that vitamin B6 (pyridoxine) supplements taken before sleep can increase dream vividness and dream recall [59]. Another study showed that vitamin B6 significantly increased the amount of dream content that participants remembered but did not significantly affect dream vividness or bizarreness, nor did it significantly affect other sleep-related variables. However, another group of the experiment that took the B complex had worse sleep quality [60]. These two studies speculate that the likely cause of B6’s impact on dreams is its role as a co-factor in the conversion of L-tryptophan to 5-HTP and the conversion of 5-HTP to 5-HT. In sleep studies, it is hypothesized that high levels of 5-HT before sleep suppress REM sleep in the early cycles, and thus induce REM sleep rebound at the end of the night, increasing and intensifying dreams. In addition, pyridoxine disulfide (called pyritinol, a derivative of B6) can be used as a nootropic. Increased levels of concentration of choline in cholinergic neurons have been observed in rats, and another study found learning and memory improvements with its administration [54]. Therefore, despite controversies with Ritcher’s patent, where vitamin B seems to be used in general, B6 might be a way to help dreamers become lucid, although more studies on the B complex related to sleep are needed. Moreover, the derivative pyritinol might be a potential candidate to induce LD, but further studies should provide more evidence of its mechanism of action.

## 4. Plants with the Potential to Intensify Oneiric Activity

Humans have always used natural resources for the maintenance of life. Plants and herbs are used as traditional medicine through the extraction of specific molecules with therapeutic potential. On this topic, we first present plants documented culturally as intensifiers of lucid and non-lucid dreams, as well as cognition. Then, we review Ayurvedic medicine and nootropic substances that modulate oneiric activity.

### 4.1. Papaver somniferum

Sleep-inducing drugs were empirically administered in ancient times for different psychiatric diseases recognized as paranoia, and drugs with hypnotic effects were classified as temperature reducers [61]. One of these sleeping-inducing drugs was poppy (*Papaver somniferum*, Papaveraceae), the plant used in the production of opium. There are several lines of evidence of opium and poppy use in the ancient era. The use of opium poppy as a treatment for children’s insomnia is described in the Egyptian Papyrus of Ebers dated to the 16th century BCE [62] and was later cited in books reporting the medicines used by Paracelsus, Galen, and Avicenna, among other important physicians and naturalists [62,63]. Greco-Roman physicians knew the dangers and benefits of the opium poppy and that greater dosages were required to obtain its pleasure effects, including sleep accompanied by alluring dreams and visions [64,65].

Several poppy-derived products have been used over time, such as raw opium, laudanum, laudanum tinctures, and poppy tea, among others. There are dozens of alkaloids in opium, including morphine, the most important narcotic substance of poppy. Morphine was first isolated in 1806 by Friedrich Sertürner, which demonstrated that it was the sleep-inducing narcotic substance of opium [63]. Sertürner first referred to the substance as the “*principium somniferum*” and later named it “morphium” after Morpheus, the Greek god of dreams. Later, the alkaloid was renamed “morphine” by Joseph Louis Gay-Lussac [63]. There are several representations of the narcotic properties of morphine and poppies in mythology. According to Greek mythology, Hypnos and Thanatos are the personifications of sleep and death and are usually associated with opium poppies [63]. A marble sculpture of Bertel Thorvaldsen of 1815 represents the Day and Night angels; Night is represented holding two children, sleep and death, and Night has her hair decorated with poppy branches [65] (Figure 4). Despite being a natural alkaloid, it is accepted nowadays that morphine is synthesized by mammalian cells from DA, although the function of endogenous morphine in the body is still a matter of debate [66].

### 4.2. Calea ternifolia

*Calea ternifolia* (also known as *Calea zacatechichi*), pertaining to the Asteraceae family and known as “the dream herb”, is an endemic species from Central America, traditionally used by the Chontal people from Mexico for divination due to its properties that increase dreaming. Mayagoitia et al. [67] reported some of its characteristics. The plant was collected and extracts were made with the supervision of the Chontal folk. Both extracts increased light sleep (N2 stage), the spontaneity of waking up during a nap period, and vividly hypnagogic imagery. The spontaneity of waking imagination also suggests hypnopompic imagery, and both of these states are used for inducing LD. Less deep sleep (N3 stage) and REM sleep was also observed, the latter contrasting with the literature that dreams are more associated with REM sleep, making way for dreams in other sleep stages, since dream recall was also increased by the extracts. It was also observed that methanol extract was associated with more recalled dreams, less content, and more vivid dreams. This larger effect of the extract suggests that the active compounds might be present in the polar fraction of the substance. *Calea* induced a discrete increase in all sensorial perceptions, discontinuity in thoughts, a rapid flux of ideas with difficulty in their retrieval, and statistically significant slowness of reaction time, which might have also induced a light hypnotic state [67,68].

One of the mechanisms of action of *Calea* is through its sesquiterpene lactones. A recent review found 37 sesquiterpene lactones in this plant [69]. There are not many reports of extracts of plants that have sesquiterpenes with AChEI activity compared to alkaloids, such as galantamine and huperzine A, but data indicate this type of activity mentioned in some plants and their respective sesquiterpenes. Recently, sesquiterpenes have been proposed for the treatment of Alzheimer’s disease due to their AChEIs properties with much therapeutic potential, mainly lactone ones, which might inhibit AChE more than galantamine [70]. In this manner, *Calea* might be able to induce LD, possibly due to its AChEI properties.

### 4.3. Celastrus paniculatus

Another sesquiterpene mentioned by Arya et al. [70] is Dihidro-β-agarofurano found in the Celastraceae family, like in *Celastrus paniculatus*, which when administered in C. elegans had rejuvenating properties [71]. This plant has been used for more than a thousand years in Indian Ayurvedic medicine, commonly called “the intellect tree”, from Hindu, Malkangani. *Celastrus paniculatus* has therapeutic potential, from oil and methanol extract to seeds [72,73]. Its neuroprotective potential has already been tested for schizophrenia induced by ketamine in rats, treating the disorder combined with clozapine or alone. Two weeks of *Celastrus paniculatus* treatment restored dendritic atrophy in the hippocampus and synaptic plasticity, also reducing AChE in the frontal cortex, hippocampus, and hypothalamus in stressed rats. Additionally, memory and cognition were augmented by *Celastrus*, possibly due to AChEI activity. In vivo studies showed restoration of working memory and spatial learning [72]. *Celastrus* was also observed to be nontoxic in the doses used, demonstrating its possible path for clinical studies in humans. Hence, it could be used for the induction of LD.

### 4.4. Silene capensis

Used by the Xhosa people from South Africa, *Silene capensis* is called the “African dream root”. Hirst [74] presents Xhosa dream reports and discusses a medical category of plants that enhance dreams (ubulawu) through his ethnographic study. A beverage is prepared by the root being ground and shaken in a beaker of water until it produces a thick white foam, which novice diviners consume on an empty stomach to increase dreams. The goal is to fill the stomach with foam until regurgitation, which indicates that a sufficient amount has been consumed. Novices consume and wash themselves with the residues during a period of three days of the full moon. The root is also chewed. This plant is used for communication with ancestors through dreams, people with trouble remembering dreams, and to cause more vivid and memorable dreams. Evidence points out that the ubulawu root has principal effects on lucid and prophetic dreams through its triterpenoid saponins [74], which are known to have AChEIs properties [75]. Other plants from the same genus have been reported with these properties [76,77,78]. In addition, the family that these plants compose, Carophyllaceae, are known to produce foam, from saponins. These plants are used for the preparation of the psychoactive drink ubulawu, traditionally used in Southern Africa, in spiritual healing processes [79]. It is feasible that the plant has potential for LD induction, also by inhibiting AChE.

### 4.5. Artemisia vulgaris

Another common herb used for intensifying dreams is Mugwort (*Artemisia vulgaris*), like *Calea zacatechichi*, also from the Asteraceae family. By inhibiting the degradation of serotonin by the monoamine oxidase (MAO) enzyme, the herb has already been reported to cause vivid dreams, where the REM sleep rebound at the end of the night may contribute to this experience. It is also observed that some lactone sesquiterpenes in this plant might also have MAO inhibition properties [80]. It is noted that the herb is also an antioxidant—like many of the herbs reported here as well as galantamine—which supports the idea that Artemisia might help to trigger lucidity during dreams.

### 4.6. Withania somnifera

Traditionally used in Ayurvedic medicine, the herb *Withania somnifera,* commonly known as Ashwagandha, is classified in the Solanaceae family. Major bioactive compounds are steroidal lactones, called withanolides, which are potent antioxidants together with alkaloids. Restoring spatial memory, motor learning, muscarinic receptor activation, and oxidative stress are cited in Bashir and colleague’s recent review [81]. They also cite anti-Alzheimer properties—inhibition of AChE and enhanced choline acetyltransferase level in rats—which are an effect of the active compound whitaferin-A (withanolide). Increased cholinergic transmission in the basal ganglia and cerebral cortex might explain the cognition improvements induced by these substances [81]. The ethanolic root extract improved gripping ability and motor movements, as well as increasing striatum DA in rats. Another ethanolic root extract had neuroprotective effects on nigrostriatal dopaminergic neurons [81].

In vitro screening of the extract was found to protect against acrolein-induced toxicity and, henceforth, could have a neuroprotective effect on patients with Alzheimer’s disease, since acrolein is found to be significantly increased in these patients [82]. A mouse model of Parkinson’s disease treated with *Withania somnifera* was found to inhibit the oxidative stress and apoptotic pathways of dopaminergic neurons (which could be related to pathways in basal ganglia, especially the substantia nigra). Improvement in motor deficits and enhanced quality of walking was observed [83]. Taking these results, AChEI activity, as already cited in many cases during this review, might be indicative that the plant can be used to induce LD.

A recent review of the effects of this herb on sleep included five randomized controlled trials, with healthy volunteers, stressed adults, or those with insomnia. In all studies, Ashwagandha was found to be beneficial and induced significant improvement in overall sleep compared to placebo groups. In addition, mental alertness was better, and reduced anxiety was observed [84]. Ashwagandha root extract was found to improve recall memory and focus, lower serum cortisol, improve sleep quality, and lower stress in a final sample of 125 individuals who took the extract (or placebo) for 90 consecutive days. Importantly, in this double-blind, randomized, parallel-group, two-arm, placebo-controlled trial, Ashwagandha was well tolerated [85]. Mild cognitive impairment has been ameliorated by Ashwagandha extract in a prospective, double-blind, placebo-controlled clinical study [86]. Task results can be summarized as improved memory, such as for faces and family pictures, as well as general and logical memory. Executive functions, including working memory, attention, and information processing speed, were improved compared to the placebo group.

The aforementioned results might favor *Withania somnifera* extract as a potential substance to induce LD. Importantly, it seems to be well tolerated in the general population. Unfortunately, we could not find any scientific studies of the plant being used to induce LD or to have oneiric properties. Moreover, sleep stages were not mentioned to have been affected in a recent review cited previously [84]. Even with no scientific reports, the molecular targets, as well as enhancing cognition, support the notion that this plant could be used for potentiating the probability of achieving lucidity in dreams.

### 4.7. Nootropics and Ayurvedic Medicine

Nootropics are a group of substances that might enhance cognitive function and can be further added to the list of candidates to induce LD. Some nootropics are found in the Ayurvedic system of knowledge too. Their mechanism of action, in general, is through improving glucose and oxygen brain supply [54]; henceforth, they might have antioxidant effects—such as many of the substances cited during this review. Thus, they might be efficient in improving cognition, such as combating oxidative stress in Alzheimer’s disease. Malík and Tlustoš [54] cite other substances that can be regarded not just as cognitive enhancers during the day, but also during night sleeping. Meclofexatone is a combination of DMAE (Deanol) and synthetic auxin, and it seems to be twice as effective as DMAE regarding its effects on choline and ACh levels in the brain. This substance normalizes oxidative stress in rats, and increased mental alertness was observed in humans.

*Ginkgo biloba* is a plant with antioxidant properties due to triterpene lactone compounds, and it also has cholinergic mechanisms. It seems that, for the treatment of Alzheimer’s disease, *Ginkgo* combined with donepezil, in contrast to donepezil alone, is superior in terms of safety and efficacy [87]. Human studies have shown improved working memory and information processing speed. Asiatic Pennywort (*Centella asiatica*), used in traditional medicine, has anti-Alzheimer’s disease properties, and its main active compounds are triterpenoids, especially triterpene saponins such as those that might be contained in *Silene capensis*, which might have properties for increasing ACh synthesis, improving memory and cognition. In rats, learning and memory were improved through modulating DA, 5HT, and noradrenaline (NA) systems with an aqueous leaf extract [88]. The eclipta species (from the same family as *Artemisia* and *Calea*—Asteraceae) has its major compounds as alkaloids, triterpenoid saponins, volatile oil, sterols, and flavonoids. Studies have shown improved learning and memory abilities in rats. Butanol fractions increase ACh and reduce oxidative stress. Another interesting plant, Water Hyssop (*Bacopa monnieri*), which can be used in cooking, has effects on enhancing attention, cognition, and learning, restoring cognitive dysfunction possibly by increasing ACh in mice brains. *Convolvulus pluricaulis,* used in Ayurveda, seems to improve memory and cognition, as well as having AChEI properties [54,88].

Overall, these cited substances seem to have key properties found in many drugs that induce LD—enhancing memory, learning, cognition, working memory, and attention, as well as antioxidative and ACh-increasing properties—especially those with AChEI activity. Indian herbal formulations that have been studied in Alzheimer’s disease might be promising in potentiating the possibility of achieving lucidity in dreams. These drugs have many types of plants in their constitution; for simplification, we will only cite the plants that are presented in this review, and for further investigation, we suggest seeing the review by Mehla et al. [88]. Improvements in learning, memory, and cognition, as well as AChEI activity, have been found in these compounds. Such compounds, even though they are not fully understood on a molecular level, might have properties for inducing LD.

## 5. General Discussion of the Neuropsychopharmacological Induction of Lucid Dreams

This review collected a wide diversity of articles that enabled a better understanding of the neurophysiological and phenomenological aspects of LD, thus promoting a comprehension of the use of exogenous substances to induce LD. For a summary of the studies that report the use of substances to induce oneiric activity, see Table 1 below.

Among the promising results, the AChEIs stand out, which are generously reported to induce LD. In addition to their scientific validation, they have already been used in various cultures. Galantamine and its influence on the dopaminergic system through the modulation of ACh can jointly generate self-monitoring phenomena, attention, cognitive clarity, recovery of memory about aspects of the self, working memory, and self-awareness. This is mainly due to the effects seen in the frontal lobe (especially the prefrontal cortex), hippocampus, and precuneus, highlighting the working memory modulated by both the cholinergic and dopaminergic systems. Based on this medicament, hybrid molecules can be created that are less toxic and inhibit AChE more than galantamine itself [24]. Therefore, it is a possibility to synthesize AChEIs for the induction of LD.

Although there are no reports of its potential related to dreams, *Celastrus paniculatus* is a strong candidate for LD induction, with AChEI mechanisms and modulating aspects similar to galantamine. An interesting mechanism is its restoration of dendritic atrophies in the hippocampus and prefrontal cortex related to depletion caused by AChE, which may be one of the mechanisms that give it the name of the “intellect tree” in the East, helping in memory and learning. Its mechanisms of action related to ACh and DA may be opportunistic in inducing LD, and studies on its therapeutic potential are well-received. *Artemisia vulgaris* and *Calea zacatechichi* could also have their studies deepened due to their AChEI potential, especially *Artemisia* due to the MAO inhibition property. The three plants mentioned in this paragraph contain sesquiterpene lactones, which should be a target in understanding the pharmacological induction of LD.

LD can be used to treat recurrent nightmares; however, in some cases, LD itself may be the symptom that needs to be treated. This is defined as a lucid nightmare, where the individual has little or no control over what happens in the dream and is consciously trapped and subjected to their fears and traumas. This might be one of the main limitations of LD therapy to recurrent nightmares [89]. From another perspective, it is observed that lucid nightmares are closely related to strong memories and traumas that have consolidated throughout an individual’s life; thus, these experiences tend to manifest more frequently in dreams. It would be interesting to know if pleasurable life experiences also tend to manifest more in dreams and help to reach the threshold that makes the dream lucid, as seems to be the case with terrifying lucid nightmares. These impactful mechanisms are mainly related to DA, possibly by activating limbic and mesolimbic areas, as well as reality self-monitoring.

Another interesting point to note is the similarities between dreams, psychoses, and psychedelic substances. These states can serve as models for specific aspects of LD [90,91,92]. The internally generated hallucinations in dreams together with the knowledge that it is a perception generated within oneself and the hallucinations induced by psychedelics may share neural substrates related to the neurotransmission of 5-HT, where the functionalities of mental and sensory images of this neurotransmitter may play an important role. In addition, substances that help improve metacognition in psychotic patients, keeping them in a state of self-awareness without fully immersive delusions, should increase the likelihood of LD. An example of this is that AChEI drugs might be useful for the treatment of psychotic symptoms. Thus, the similarities between these states can help in understanding lucidity in a more general way, not just in dreams but in understanding the state of consciousness one is in. Paradoxical pharmacological dissociations are an interesting phenomenon since drugs inducing delta activity, a state normally associated with loss of consciousness, could provide biomarkers for the neural substrates of awareness, where the subject remains conscious. Delta-enhancing drugs such as N,N-dimethyltryptamine [60] do not necessarily induce loss of consciousness, and might even enhance it in some cases, such as augmented visual imagery in some episodes of LD.

Since LD is extremely sensitive to suggestion, for future studies, pharmacological techniques could be combined with behavioral/cognitive techniques. The latter includes a variety of types of induction techniques, which can be divided into dream-initiated LD (including MILD, reality testing, and Tholey’s combined techniques) and wake-initiated LD (sense-initiated LD) (for review, see [93]). This division makes a distinction between those LD episodes that start during the dream and those that start as the subject enters consciously into sleep. Other ways to enhance the probability of becoming lucid include a dream diary and mindfulness interventions. Studies have already found a possible role of mindfulness practices and meditations in inducing LD [6,94,95,96], besides ancient techniques that cultivate self-consciousness during sleep. Further research should explore the AChEI properties of the various substances mentioned, integrating pharmacological and cognitive techniques for reliable LD induction.

## 6. Limitations, Perspectives, and Conclusions

Some limitations must be considered. Firstly, an important point to recognize in neurotransmission is the diversity and complexity of neurotransmitters, their different production sites and projection areas, and the great diversity of receptors. Each neurotransmitter can have distinct or similar functions to others, depending on these points. In addition, they modulate other neurotransmitter systems. Considering this extremely complex brain network that functions in circadian and homeostatic ways and produces the experience of human reality perception, what has been described here is a mere attempt at understanding how lucidity is achieved on a neurotransmitter level. Moreover, no studies have yet been conducted on eye movement verification related to the pharmacological induction of LD. Therefore, it would be interesting to both verify the eye movements during LD induced pharmacologically and test whether galantamine and its potential AChEIs activate areas related to self-awareness, self-monitoring, and metacognition, with emphasis on dopaminergic modulation. Furthermore, it is possible to utilize the interaction of plants with specific properties to facilitate the induction of LD. However, a comprehensive understanding is required to discern if there exists a synergy between these plant-based compounds, the potential risks associated with such synergies, and the cognitive–behavioral effects induced by these interactions. It is noteworthy that there is a scarcity of studies focusing on plants at present, which could potentially contribute to discussions advocating for streamlined regulatory processes for human trials involving plant-derived substances. This scarcity of research serves as an observable limitation in exploring the efficacy and safety of plant-based interventions for inducing LD.

This review integrated numerous studies to deepen the comprehension of LD. In light of this, we propose a research agenda to steer forthcoming investigations towards the pharmacological induction of LD. Future studies are urged to concentrate on elucidating the impact of various drugs and plants on sleep and dreaming, an area significantly underexplored in the current literature. Further details of this comprehensive research agenda are outlined below, delineating pathways for interdisciplinary collaboration and innovative methodological approaches.
Research agenda:
Future studies should focus on how the aforementioned drugs and plants have an impact on sleep and dreaming. In addition, there is a potential influence of the chronobiological effects of time on the administration of these drugs; some of them (e.g., nootropics) mostly affect the waking state, but also might impact nocturnal consciousness. The literature on this type of information is scarce.The pharmacodynamics of plants and drugs should also be researched, mostly those that have similar pharmacological, cognitive, behavioral, and neurophysiological aspects to LD.Sesquiterpenes from known plants, which possibly already have approval for human testing (to reduce bureaucracy), should be researched using Ellman’s method. This allows for the elaboration of an AChEI’s concentration and efficacy, which can be compared to medications such as galantamine and donepezil.Better understanding how sesquiterpenes influence DA and ACh concentrations in the brain would be a great step, allowing for inferring its action in specific brain areas.Pharmacological protocols for the understanding of the synergism between potential drugs/plants to induce LD is crucial. In this way, a combination can be used to enhance the probability of LD and to know whether there are side effectsDepending on the focus of future studies, many methodologies can be created. A mixture of induction techniques, including behavioral and pharmacological, should be combined to enhance the probability of achieving dream lucidity in a controlled environment. In addition, the eye-signal technique to flag LD and neuroimaging techniques could be used in conjunction.

LD is a complex state full of opportunities for the dream self. Understanding neural systems and brain areas enables the induction of this state effectively, allowing for the study of neural correlates of self-consciousness. Finally, it is important to note that to study the pharmacological induction of LD, a close interaction of biomedical research with anthropology and the history of ancient people who used oneiric plants is necessary. This collaboration of neuropsychology and social sciences can be fruitful for studies that aim to understand human consciousness.

## Figures and Tables

**Figure 1 brainsci-14-00426-f001:**
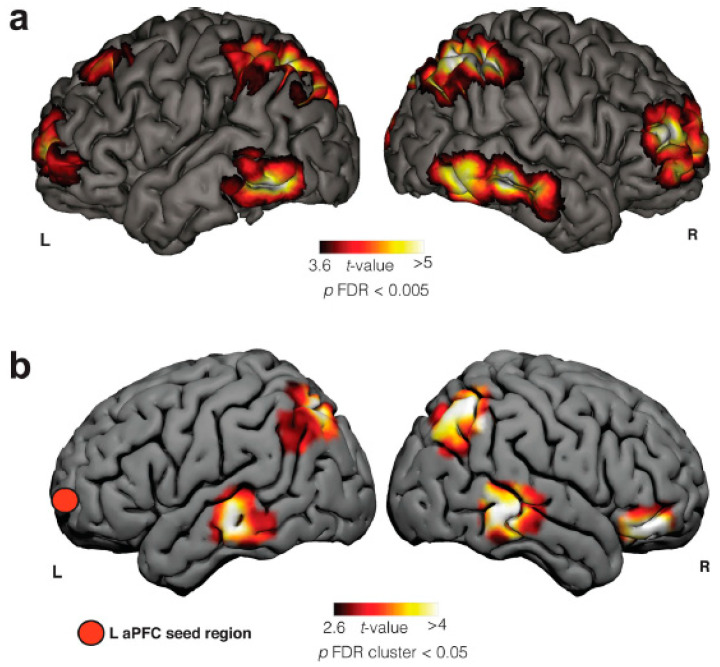
Frontal, parietal, and temporal brain areas are involved in LD, which is evidenced by two different methods of neuroimaging: (**a**) Blood-oxygen-level-dependent (BOLD) activity in an fMRI case report of LD. (**b**) Seed-based resting-state (SBRS) functional connectivity differences between frequent lucid dreamers and non-frequent lucid dreamers (control group). Adapted from [15] with permission from the authors. R = right side and L = left side. aPFC = anterior prefrontal cortex.

**Figure 2 brainsci-14-00426-f002:**
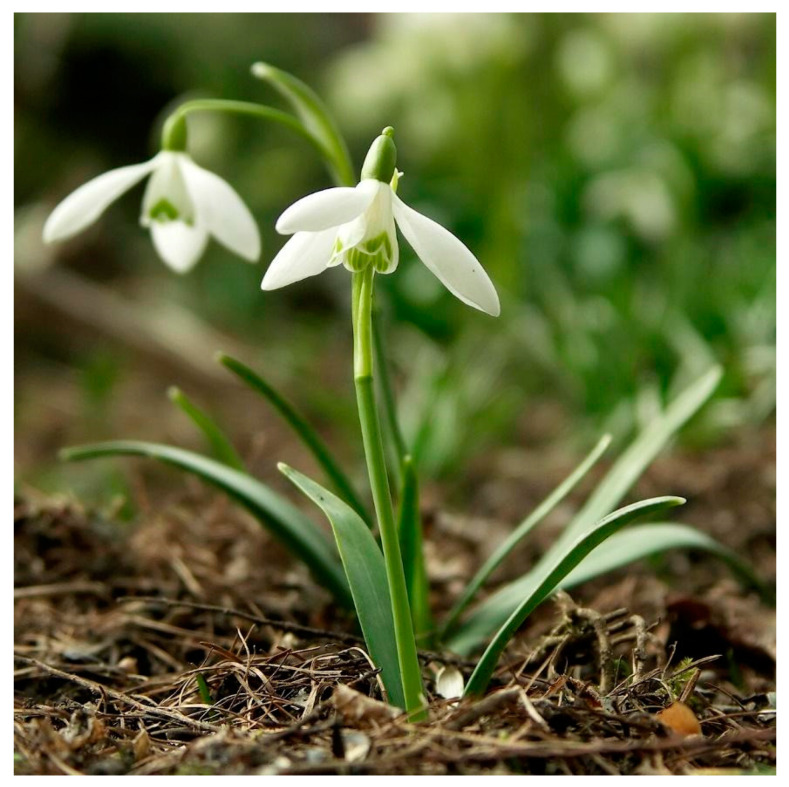
Galanthus flower, also known as Snowdrop [26].

**Figure 3 brainsci-14-00426-f003:**
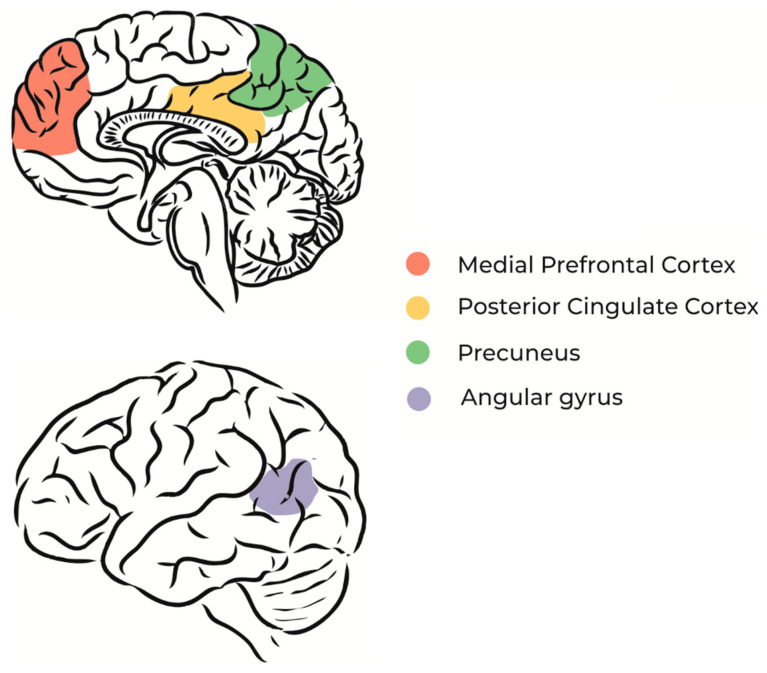
The main regions that comprise the default mode network (DMN). (**up**) Medial view: (brown) medial prefrontal cortex; (yellow) posterior cingulate cortex; (green) precuneus; (**down**) lateral view: (gray) angular gyrus. Adapted from [39] with permission from the authors.

**Figure 4 brainsci-14-00426-f004:**
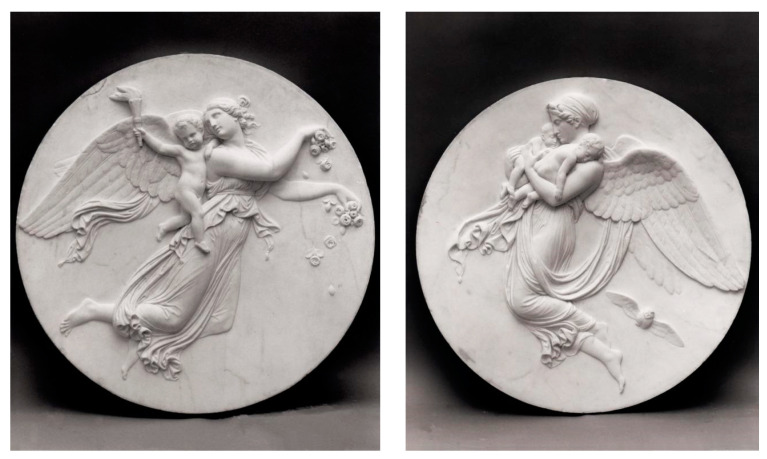
Day and Night angels, by Bertel Thorvaldsen (1815). (**left**) The Day angel. (**right**) The Night angel, with poppy branches in her hair, holds two children: Hypnos, who represents sleep, and Thanatos, who represents death.

**Table 1 brainsci-14-00426-t001:** Studies that report the use of substances to induce lucid or state-like LD.

Authors (Year)	Study Design	Substances/Techniques	Mechanism of Action	Results	Commentary
LaBerge (2004) [28]	Patent	Donepezil, Rivastigmine, Galantamine, and Huperzine	Acetylcholinesterase inhibitors (AChEIs)	Donepezil: 90% of subjects induced LD. Similar dose-dependent results were encountered for rivastigmine and galantamine, with fewer side effects. The same for huperzine, but the dosage was inconclusive.	Reports a series of methods utilizing memory enhancing drugs such as those used for Alzheimer’s disease—AChEIs. Other possible mechanisms cited: cholinergic agonists; muscarinic receptor agonists; allosteric modulators of ACh and nicotinic receptors.
LaBerge et al. (2018) [29]	Experimental study	Galantamine, WBTB, MILD, lectures about LD for recognizing dream cues	AChEI	62% reported LD: 14% on placebo; 27% with 4 mg; and 42% with 8 mg.	Galantamine induces LD in a dose-dependent manner. The integrated protocol seems to be effective in induction. Authors comment that galantamine may impact REM sleep, reducing latency and increasing phasic activity. Galantamine enhances dopaminergic neurotransmission, which might be involved in metacognition and conscious self-monitoring.
Sparrow et al. (2018) [30]	Experimental study	Galantamine, MDR, WBTB	AChEI	LD was reported by 40% of participants on WBTB + galantamine; 34% reported LD on MDR + galantamine.	MDR condition might expose traumas and conflicts, possibly for resolution.
Kern et al. (2017) [31]	Experimental study	L-alpha glycerylphosphorylcholine (α-GPC)	ACh precursor	No significant results.	No cognitive techniques were explicitly used in combination, which might impact results.
Haas et al. (2022) [50]	Case study	Pregabalin	Binding to alpha-2-delta sites, reducing excitatory neurotransmission. Anti-glutamatergic reports	Patient with multiple myeloma and history of chronic pain. Drug was administered causing lucid nightmares; after discontinuation, it ceased.	Abnormal dreams and visual hallucinations are uncommon symptoms of pregabalin.
Biehl (2022) [52]	Survey study	Vitamin intake, fish, fruit, and chili consumption, and antidepressants	Omega-3 (fish, which might contain DMAE), capsaicin (chili), antidepressants (availability of serotonin and receptors)	Significant correlations of the substances with LD.	Many of the substances affected dreaming, nightmares, LD, and hypnagogic state.
Sergio (1988) [53]	Self-case report	DMAE	Converted to choline by the body, stimulating reticular formation	Combining WBTB and visualization techniques, the author has suggested that it has great effects on inducing LD.	The lactate salt is the most effective, followed by the p-acetamidobenzoate salt, and the tartrate salt is the least effective.
Richter (2007) [55]	Patent	Primary ingredients of the supplement: Calea ternifolia, L-5-HTP, and vinpocetine	Calea ternifolia: AchEI L-5-HTP: increases serotonin Vinpocetine: anti-oxidant	The author gives two examples of people with recurrent nightmares. When taking the supplement, subjects started resolving conflicts through enhanced vividness and lucidity during dreams.	Secondary ingredient: melatonin; tertiary ingredients: wild lettuce extract, Mugwort extract (Artemisia vulgaris), DMAE, passionflower extract, and green tea extract.
Ebben et al. (2002) [59]	Experimental study	Pyridoxine (B6)	Co-factor for production of 5-HT	Increased recall and vividness in dreams.	High levels of serotonin before sleep suppress REM, causing rebound at the end of the night.
Aspy et al. (2018) [60]	Experimental study	Pyridoxine and B Complex	Co-factor for production of 5-HT (B6)	Increased oneiric content recalled for B6. B complex worsened sleep quality.	Besides serotonin, nootropics derivatives from B6, like pyritinol, might combine effects for LD induction.
Mayagoitia et al. (1986) [67]	Experimental study	Calea zacatechichi	AChEI (sesquiterpene lactones)	Individuals reported increased light sleep; vividly hypnagogic imagery; less REM and deep sleep; more recalled and vivid dreams; and less content.	Many sesquiterpene lactones have been found in this plant. The Asteraceae family, including Calea, Artemisia, and Eclipta plants, have a potential role in the modulation of the cholinergic system to induce LD.
Hirst (2005) [74]	Ethnography study	Silene capensis	Saponins—AChEI	Main effects reported by diviners and novices were lucid and prophetic dreams.	Ubulawu drinks containing medicinal plants, which some have triterpenoid saponins for AChEIs, are used in rituals for divination for medicinal purposes.

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
