# Peer review of "Neuropsychopharmacological Induction of (Lucid) Dreams: A Narrative Review"

_brainsci, 2024, doi:10.3390/brainsci14050426_

Round 1

Reviewer 1 Report

Comments and Suggestions for Authors

The manuscript provides an interesting overview of the behavioral and pharmacological induction of lucid dreams. Although this topic has recently received a review (Tan & Fan, 2023), in a very innovative way, this manuscript focuses on the possible substances/plants that can contribute to reaching lucidity in dreams. In this regard, given the very scarce evidence in the literature, the choice to conduct a narrative review (instead of a systematic one) may represent a good starting point for future research on the topic. However, I have some comments that should be addressed.

Major points:

The title can be improved by highlighting the narrative purpose of the review.

Generally, the review is quite difficult to follow. This is due to the large amount of reported information, much of which has more didactic than explanatory value. In this regard, the first two sections of the review (i.e., 1.1 and 1.2) might be inserted in the supplementary materials, resulting in a more readable manuscript. 

For the same purpose, some tables could be included to describe the studies conducted and the techniques/substances used, as well as to describe the substances and their possible effects.

Throughout, very few studies that have used a substance for the induction of lucid dreams are described, and sections often abound with additional information on the general effects of the substances. This makes the manuscript hard to follow, therefore this kind of information should be reduced or discussed in a separate section from the description of lucid dream studies.

Minor points:

I am not sure if it is possible to use figures derived from other articles, but if so, all elements of the figure should be explained (e.g. acronyms).

The quality of the review could be enhanced by discussing the possible role of behavioral techniques in combination with pharmacological techniques to induce lucid dreams.

Useful information for future studies could be provided by using a “practice point” and/or a “research agenda” section.

Author Response

Reviewer 1

1) This topic has recently received a review (Tan & Fan, 2023)

R: We added this reference (93) and a brief discussion at the last paragraph of Topic 5. General Discussion of the Neuropsychopharmacological Induction of Lucid Dreams.

2) The title can be improved by highlighting the narrative purpose of the review

R: We changed the title to: “Neuropsychopharmacological Induction of (Lucid) Dreams: a Narrative Review”. 

3) Generally, the review is quite difficult to follow. In this regard, the first two sections of the review (i.e. 1.1 and 1.2) might be inserted in the supplementary materials, resulting in a more readable manuscript. 

R: We moved sections 1.1 and 1.2 to Supplementary Materials, and added a brief Introduction in the main text.  

4) For the same purpose, some tables could be included to describe the studies conducted and the techniques/substances used, as well as to describe the substances and their possible effects.

R: We added a Table at the General discussion section, describing the main studies, substances used, type of study, mechanism(s) of action, main results and other complementary commentaries.

5) Very few studies that have used a substance for the induction of lucid dreams are described, and sections often abound with additional information on the general effects of the substances. This makes the manuscript hard to follow, therefore this kind of information should be reduced or discussed in a separate section from the description of lucid dream studies.

R: We removed all information on the general effects of the substances, to focus on their effects on lucid dreams. 

6) I am not sure if it is possible to use figures derived from other articles, but if so, all elements of the figure should be explained (e.g. acronyms)

R: We asked for the authors permission to use Figures 1 and 3, and they authorized. Figures 2 and 4 do not need copyright permission. Besides, we removed unnecessary acronyms and explained the important ones in the new legends.  

7) The quality of the review could be enhanced by discussing the possible role of behavioral techniques in combination with pharmacological techniques to induce lucid dreams.

R: We added a discussion about the combination of behavioral and pharmacological techniques to induce lucid dreaming at the last paragraph of Topic 5. General Discussion of the Neuropsychopharmacological Induction of Lucid Dreams.

8) Useful information for future studies could be provided by using a “practice point” and/or a “research agenda” section.

R: We added this section describing useful information for future research at the end of section 6. Limitations, Perspectives, and Conclusions.

Reviewer 2 Report

Comments and Suggestions for Authors

This manuscript is written logically, being well structured and easy to read.

The methods are relevant and well selected

I did not notice any grammatical or spelling mistakes.

General comment:

In this study, the authors aimed to review the literature on the neuropsychopharmacological induction of lucid dreaming (LD), a physiological state of consciousness in which dreamers become aware that they are dreaming and can control the content of the dream. LD is rare in the general population, which has led to increased interest in methods of its induction.

The review begins by describing the circadian and homeostatic processes involved in sleep regulation, as well as the mechanisms controlling rapid eye movement (REM) sleep, with an emphasis on neurotransmission systems. The neurophysiology and phenomenology of LD are then discussed to understand the cortical oscillations and brain areas involved in the emergence of lucidity during REM sleep.

Subsequently, the review explores possible exogenous substances, including natural herbs and artificial drugs, that have the potential to increase metacognition, REM sleep, and/or dream recall so that they can induce LD. The main focus is on substances that increase cholinergic and/or dopaminergic transmission, such as galantamine.

However, the complexity of these neurotransmission systems presents a challenge in interpreting the results. Despite promising candidates such as galantamine, the review concludes that more research is needed to identify reliable methods of inducing LD by pharmacological means. The subject is an exciting and interesting one.

I recommend increasing the quality of figure 1. Currently the writing is very difficult to understand.

Author Response

We are grateful to the anonymous reviewer for the positive appraisal. We addressed the concerns raised, and updated it in the revised manuscript.

Reviewer 3 Report

Comments and Suggestions for Authors

This paper provides a review of the literature on neuropsychopharmacological induction of lucid dreams, and emphasizes circadian and homeostatic processes of sleep regulation and the mechanisms that control rapid-eye-movement sleep with a focus on neurotransmission systems.

I believe, this review can be recommended for publishing. There are several points for improvement:

  1. At P.2, L.79 the authors refer to “ultradian oscillator”. Using such term suppose defining some mathematical characteristics for sustained rhythmicity – exact frequencies/period(s) range, and phase stability. Excitatory role of REM-on (cholinergic) cell clusters s and inhibitory role of REM-off (aminergic) cell clusters is considered and activation of brain regions is discussed. But do we know, what initially drives such oscillations, and what brain structure can be viewed as central for sustaining such oscillations, similarly to suprachiasmatic nucleus as master clock cite of the circadian oscillations?

  2. Another suggestion concerns consideration of substances with oneiric activity and nootropic activity. One may wonder how much neuropsychopharmacological effects after administered of these substances may vary around 24-hour time scale in context of the featured review, considering that effects first group (oneric) can be more effective when administered before sleep, while for nootropic substances primary effect can be in the morning but also affect dreaming.

  3. The authors may wish to give a bit more information on melatonin, which is mentioned twice, but its specific role in coordination of sleep and sleep stages was not yet clarified in this review.

Author Response

Reviewer 3

1) At P.2, L.79 the authors refer to “ultradian oscillator”. Using such term suppose defining some mathematical characteristics for sustained rhythmicity – exact frequencies/period(s) range, and phase stability. Excitatory role of REM-on (cholinergic) cell clusters and inhibitory role of REM-off (aminergic) cell clusters is considered and activation of brain regions is discussed. But do we know, what initially drives such oscillations, and what brain structure can be viewed as central for sustaining such oscillations, similarly to suprachiasmatic nucleus as master clock cite of the circadian oscillations?

R: To avoid confusion, and since chronobiological factors are not the center of our manuscript, we removed the term “ultradian oscillator”. This section can now be found in Supplementary materials, as suggested by Reviewer 1.  

2) Another suggestion concerns consideration of substances with oneiric activity and nootropic activity. One may wonder how much neuropsychopharmacological effects after administered of these substances may vary around 24-hour time scale in context of the featured review, considering that effects first group (oneric) can be more effective when administered before sleep, while for nootropic substances primary effect can be in the morning but also affect dreaming.

R: The authors agree with the reviewer regarding the potential influence of chronobiological factors on the effects of substances, such as nootropics and/or oneiric compounds. However, due to the scarcity of studies exploring or acknowledging the timing of administration, and considering that our manuscript primarily focuses on other aspects, we have acknowledged this as a crucial area within a new text regarding Research Agenda for future studies, at the end of topic 6. Limitations, Perspectives, and Conclusions.

3) The authors may wish to give a bit more information on melatonin, which is mentioned twice, but its specific role in coordination of sleep and sleep stages was not yet clarified in this review.

R: We added a discussion about melatonin at subsection 3.3 Case Reports with Other Substances.

Round 2

Reviewer 1 Report

Comments and Suggestions for Authors

The authors responded to my requests. I have no further comments.